# Hazard Assessment of Polystyrene Nanoplastics in Primary Human Nasal Epithelial Cells, Focusing on the Autophagic Effects

**DOI:** 10.3390/biom13020220

**Published:** 2023-01-23

**Authors:** Balasubramanyam Annangi, Aliro Villacorta, Montserrat López-Mesas, Victor Fuentes-Cebrian, Ricard Marcos, Alba Hernández

**Affiliations:** 1Group of Mutagenesis, Department of Genetics and Microbiology, Faculty of Biosciences, Universitat Autònoma de Barcelona, 08193 Cerdanyola del Vallès, Spain; 2Facultad de Recursos Naturales Renovables, Universidad Arturo Prat, Iquique 1111100, Chile; 3GTS-UAB Research Group, Department of Chemistry, Faculty of Science, Universitat Autònoma de Barcelona, 08193 Cerdanyola del Vallès, Spain

**Keywords:** nanoplastics, polystyrene, HNEpCs, uptake, oxidative stress, mitochondrial membrane potential, autophagy

## Abstract

The human health risks posed by micro/nanoplastics (MNPLs), as emerging pollutants of environmental/health concern, need to be urgently addressed as part of a needed hazard assessment. The routes of MNPL exposure in humans could mainly come from oral, inhalation, or dermal means. Among them, inhalation exposure to MNPLs is the least studied area, even though their widespread presence in the air is dramatically increasing. In this context, this study focused on the potential hazard of polystyrene nanoplastics (PSNPLs with sizes 50 and 500 nm) in human primary nasal epithelial cells (HNEpCs), with the first line of cells acting as a physical and immune barrier in the respiratory system. Primarily, cellular internalization was evaluated by utilizing laboratory-labeled fluorescence PSNPLs with iDye, a commercial, pink-colored dye, using confocal microscopy, and found PSNPLs to be significantly internalized by HNEpCs. After, various cellular effects, such as the induction of intracellular reactive oxygen species (iROS), the loss of mitochondrial membrane potential (MMP), and the modulation of the autophagy pathway in the form of the accumulation of autophagosomes (LC3-II) and p62 markers (a ubiquitin involved in the clearance of cell debris), were evaluated after cell exposure. The data demonstrated significant increases in iROS, a decrease in MMP, as well as a greater accumulation of LC3-II and p62 in the presence of PSNPLs. Notably, the autophagic effects did indicate the implications of PSNPLs in defective or insufficient autophagy. This is the first study showing the autophagy pathway as a possible target for PSNPL-induced adverse effects in HNEpCs. When taken together, this study proved the cellular effects of PSNPLs in HNEpCs and adds value to the existing studies as a part of the respiratory risk assessment of MNPLs.

## 1. Introduction

Microplastics and nanoplastics (MNPLs) are considered emergent environmental pollutants. As such, further studies are required to understand if their exposure supposes a real health concern. The increased presence of environmental MNPLs is the obvious consequence of the increasing use of plastic goods and the poor capacity to recycle them, mainly regarding one-use plastics. The steadily increased presence of plastic waste in the environment has generated environmental alarm [1], but their easy internalization by organisms and their demonstrated cell uptake has moved the focus toward their potential health hazard [2,3,4].

The alarm for the harmful consequences of MNPLs was initiated in the marine environment. The wide dispersion of plastic waste in many coastal areas and beaches and mainly in the plastic island trapped in the oceanic gyres was a warning signal of their harmful consequences [5]. The presence of MNPLs in marine organisms and their potential amplification by the trophic web suggested a risk for humans related to the consumption of seafood [6]. Consequently, MNPL exposure via ingestion was the first approach used to determine the harmful health effects of MNPL exposure. Many studies have focused on using the intestinal barrier as a model of MNPL targets both within in vivo [7] as well as in vitro models of the intestinal barrier [8,9]. Nevertheless, soon came a conviction that MNPL exposure was a global threat [10], and MNPL presence was found in every environmental niche, including air [11,12,13].

The low weight associated with MNPLs makes their atmospheric transport and their deposition in any part of the world easy [14]. The sources of atmospheric MNPLs can be varied and in addition to the secondary MNPLs resulting from general plastic waste degradation, other sources can be relevant in specific sites like in the big cities where MNPLs resulting from tire degradation represent an important source of MNPLs [15]. Alternatively, high levels of MNPLs have also been identified in indoor environments, mainly due to textile fibers [16]. Although there are few toxicological studies evaluating the hazard effects of this via inhalation, in a study using Sprague-Dawley rats exposed to polystyrene MNPLs for 14 days, mainly molecular changes were observed, such as the increased expression of inflammatory proteins (TGF-β and TNF-α) in the lung tissue [17]. Furthermore, in a preliminary study on the effects of tire MNPLs, their inhalation by C57BL/6 mice resulted in restricted ventilatory dysfunction and fibrotic pathological changes [18]. Consequently, it seems appropriate to further analyze the harmful effects associated with MNPL exposure via inhalation on respiratory tract targets.

In vitro studies can be a good alternative to the use of mammal models, but the selection of the cell type is a crucial point, specifically in cells representative of the respiratory tract. Although several cell lines of pulmonary origin exist and have been used to evaluate MNPL effects, such as the nontumorigenic lung epithelial BEAS-2B cells [19] and lung carcinoma epithelial A549 cells [20], the use of primary cells can propose many advantages, like those from the nasal epithelium. The nasal epithelium is a physical and immune system barrier and constitutes the first structure of the respiratory system to contact with environmental pollutants. Cells are involved in the nasal mucociliary clearance that prevents such pollutants from reaching the lungs. Recently, a study has been carried out with these primary human nasal epithelial cells (HNEpCs) to determine the harmful effects of MNPLs, such as PSNPLs [21], but they have also shown sensitivity to the effects of air PM2.5 components [22]. Interestingly, a recent paper has shown that HNEpCs are much more sensitive than other cells from different parts of the respiratory tract (human bronchial epithelial cells, normal nasopharyngeal epithelial cells, human lung epithelial cells, and human lung fibroblasts) to the effects of flame retardants [23]. It must be noted that flame retardants are among the different additives added during plastic production [24].

Hence, the potential interactions between the nasal epithelial cells and MNPLs present in the air would possibly have biological consequences. In the present study, we focused on the autophagy pathway, which is activated when the cells are under duress for turning over or recycling the dysfunctional organelles and damaged proteins into simpler forms for cell survival [25]. The implications of engineered nanomaterials, having similarities to MNPLs, in the induction and blockage of autophagy as a possible toxic mechanism [26] could be taken as a cue for MNPL-induced autophagic effects. To our knowledge, only one study has been reported that deals with HNEpC and PSNPL toxicity effects [21], but not in the autophagy pathway.

In summary, following our studies on the potential hazard posed by MNPL exposure, we have used primary human nasal epithelial cells to determine the harmful effects of two different-sized PSNPLs. We chose two different sizes of PSNPLs to be representative of NPLs, as there might exist size-dependent adverse effects since PS-50 and PS-500 vary significantly in their size. Nevertheless, we are aware that this is not a complete size-dependent study since only two sizes were evaluated. In support, two recent studies showed that there were size-dependent adverse effects of PSNPLs on different cell types, such as human colon adenocarcinoma Caco-2 cells and human umbilical vein endothelial cells (HUVEC) [27,28]. Among the different evaluated effects, those affecting mitochondrial functionality and the autophagy pathway stand out.

## 2. Materials and Methods

### 2.1. Characterization of Polystyrene Nanoplastics (PSNPLs)

As we were interested in the effects of size, two pristine PSNPLs (50 and 500 nm) were selected. They were commercially purchased from Spherotech (Chicago, IL, USA) (PS-50, catalog no. PP-008-10; PS-500; catalog no. PP-05-10). In addition, PS-50 and PS-500 were labeled using the textile dye iDye Poly Pink (Jacquard Products, Healdsburg, CA, USA) for studying cellular internalization under confocal microscopy, adapting the previously reported protocols used for micro and nanoplastics [29,30]. The dispersions of the PSNPLs were made at 100 μg/mL in both Milli-Q distilled water and in airway epithelial cell growth media without serum for the physicochemical characterization studies [2].

PS-50 and PS-500 were characterized for their sizes and shapes using transmission electron microscopy (TEM, JEOL JEM-1400 instrument, Jeol LTD, Tokyo, Japan). In addition, the hydrodynamic behavior, zeta potential (as surface charge), and polydispersity index (PDI) were measured utilizing dynamic light scattering (DLS) and laser Doppler velocimetry (LDV) methodologies at the Catalan Institute of Nanoscience and Nanotechnology on a Z-Sizer (Malvern Ultra-red from Malvern Panalytical, Cambridge, UK), respectively. Finally, the chemical nature of the PSNPLs was achieved by characterizing their functional groups by Fourier-transform infrared spectroscopy (FTIR) using a Hyperion 2000 micro-spectrometer (Bruker, Billerica, MA, USA).

### 2.2. Cell Culture

The different components for the cell culture were purchased from PromoCell (GmbH, Heidelberg Germany). They included 1 vial of 5 × 10^5^ cells/mL cryopreserved primary human nasal epithelial cells (HNEpCs, catalog no. C-12620), the airway epithelial cell growth media (AEGM) kit with supplemental mix (catalog no. C-21160), and the DetachKit containing HEPES-buffered saline solution, trypsin/EDTA, trypsin neutralizing solution (catalog number C-41200). The cells were immediately thawed in a water bath at 37 °C, and subsequently transferred to prewarmed AEGM and maintained in a humidified atmosphere of 5% CO_2_ and 95% air at 37 °C. The cells were subcultured using the DetachKit at 70–90% confluency and seeded for the different kinds of experiments planned in the study. The selection of HNEpCs was based on their role as a first barrier and contact in nasal epithelium against environmental air pollutants, such as pathogens, allergens, and particulate matter (which can include environmental nanoplastics). Moreover, they could be involved in the mucociliary clearance of the nasal cavity to eliminate foreign particles [31,32].

### 2.3. Cell Viability

The viability of cells after PSNPL exposure was measured using the Beckman counter method with a ZTM Series Coulter counter (Beckman Coulter, Brea, CA, USA). To such end, 1 × 10^5^ HNEpCs were seeded in 6-well plates in triplicate and incubated overnight at 5% CO_2_ and 37 °C. The following day, the cells were treated with different concentrations of both PSNPLs (PS-50 and PS-500), ranging from 0.50 to 100 μg/mL for 24 h. The percentage decrease in cell viability after 24 h of exposure, as compared to the untreated control, was calculated by averaging three independent viability experiments.

### 2.4. Intracellular Uptake by Confocal Microscopy

The cell uptake of PS-50 and PS-500 was determined by using confocal microscopy. For determining the intracellular localization of PS-500, the particles were previously stained with the textile dye iDye Poly Pink (iDye) using an adaptation of the described protocol [29]. Briefly, 1 mL of PS-500 at the concentration of 5 mg/mL was put into a 1.5 mL tube containing 0.01 g of the dye. The mixture was vigorously vortexed and incubated for 2 h at 70 °C. After cooling at room temperature, the suspension was added to 9 mL of Milli-Q water in an Amicon Ultra-15 Centrifugal Filter Unit (Merk, Darmstadt, Germany) and centrifuged at 4000 rpm for 15 min. This step was performed twice to remove the excess of iDye. The washed particles were then collected and suspended in a final volume of 1 mL of Milli-Q water. Here on, these stained PSNPLs are quoted as iDyePS-50 and iDyePS-500.

For confocal determination, 0.3 × 10^5^ HNEpCs/well were seeded in a μ-slide 8-well Glass Bottom cell culture chamber (ibid GmbH, Gräfelfing, Germany) and treated with iDyePS-50 or iDyePS-500 at the concentration of 100 µg/mL for a period of 24 h. After that, cell samples were washed with fresh AGEM, and the nuclei of the cells were stained for 5 min at room temperature with 1:500 Hoechst 33,342 (ThermoFisher Scientific, Carlsbad, CA, USA), while the cell membranes were stained with 1:500 Cellmask™ Deep Red plasma (ThermoFisher Scientific, Carlsbad, CA, USA). The iDyePS-50 and the iDyePS-500 were visualized intracellularly at emission wavelengths of 585 nm under a Leica TCS SP5 confocal microscope. The Imaris 9.5 software was used for image processing. The images were recorded by selecting two or more random fields per sample. The experiment was performed twice independently.

### 2.5. Estimation of Intracellular ROS Levels by Using the DCFH-DA Assay

The induction of intracellular reactive oxygen species (iROS) by PS-50 and PS-500 was quantified by using the dichlorodihydrofluorescein diacetate assay (DCFH-DA, Sigma Aldrich). To proceed, 5 × 10^5^ cells/mL of HNEpCs were seeded in a 6-well plate, incubated overnight, and treated with 100 µg/mL of both PSNPLs for 24 h. The following day, the cells were incubated with 5 μM DCFH-DA in serum-free AGEM for 30 min at 37 °C [2]. After the DCFH-DA treatment, cells were washed with 1X PBS, trypsinized, centrifuged, and the pellet was resuspended to 1 × 10^6^ cells/mL in 1X PBS. Cells were immediately analyzed by using a CytoFlex flow cytometer (Beckman Coulter, Brea, CA, USA). Hydrogen peroxide (H_2_O_2_, 0.5 μM) was used as a positive control. Several groups of 20,000 cells were scored and evaluated using the CytoFlex software. A minimum of three independent experiments were carried out.

### 2.6. Mitochondrial Membrane Potential Assay

The determination of the mitochondrial membrane potential in HNEpCs due to PS-50 and PS-500 exposure was accomplished by using the MitoProbe™ TMRM kit for flow cytometry (Invitrogen, catalog no. M20036). To proceed, 5 × 10^5^ cells/mL of HNEpCs plated in 6-well dishes overnight were exposed to 100 µg/mL of PS-50 and PS-500 for 24 h. After, the cells were treated with 1 μL (final concentration 20 nM) of the cationic and lipophilic fluorescent dye tetramethyl rhodamine methyl ester (TMRM) for 30 min at 37 °C and 5% CO_2_. Subsequently, the cells were analyzed in a CytoFlex flow cytometer (Beckman Coulter, USA) with 561-nm excitation using emission filters appropriate for R-phycoerythrin. As a positive control, 1 μL of 5 mM CCCP (carbonyl cyanide 3-chlorophenylhydrazone) was applied to the cells for 5 min at 37 °C, with 5% CO_2_. A minimum of 10,000 single cells were scored per sample and were analyzed using the CytoFlex software. A minimum of three independent experiments were performed.

### 2.7. Total Protein Extraction and Western Blot Analysis

The expression of autophagy-related proteins, such as LC3-II and p62 (for induction and blockage, respectively), was assessed by Western blot once HNEpCs were exposed for 24 h to PS-50 and PS-500. The exposed cells were homogenized in NP-40 lysis buffer (composition for 500 µL per sample: 1 M Tris, 1.5 M NaCl, 20% glycerol, 0.1% Triton X 100, 0.5% NP-40, 2.5X proteases, and 0.2X phosphatases). From the obtained protein lysate, 50 µg of protein was run on an 8% (for p62) and 15% (for LC3-II) SDS-PAGE gel and transferred to PVDF membranes. After the transfer of proteins, the membranes were blocked using 5% nonfat milk for over 1 h at room temperature, followed by the incubation of membranes overnight at 4 °C with primary antibodies against LC3-II (anti-LC3A/B 12741, 1:1000, Cell Signaling Technology, Danvers, MA, USA), p62 (M162-3, 1:1000, Medical & Biological Laboratories Co., Ltd., Nagoya, Japan), and GAPDH (sc-32233, 1:5000, Santa Cruz, UK) as the loading control. Next, they were washed thrice with TBST for 5 min each and incubated with HRP-conjugated secondary antibody (at a 1:2500 dilution) for 1 h at room temperature. After that, they have washed again thrice with TBST for 5 min each and developed using an enhanced chemiluminescence system (ECL, Cell Signaling Technologies). The relative quantification of protein expression was determined by using ImageJ analysis.

### 2.8. Immunocytochemistry

For LC3II and p62 localization, the cells were exposed to iDyePS-50 and iDyePS-500 (100 µg/mL) for 24 h for immunostaining analysis. Later on, the cells were fixed in ice-cold 4% PFA for 10 min, permeabilized with 0.1% Triton X-100 (Sigma, T8787) for 15 min at room temperature, and subsequently blocked with 2% BSA for 30 min at room temperature, and incubated with anti-LC3II (2775, 1:500, Cell Signaling Technology, Danvers, MA, USA) and anti-p62 (M162-3, 1:200, Medical & Biological Laboratories Co., Ltd., Nagoya, Japan) antibodies at 4 °C overnight. The following day, the cells were labeled with secondary antibodies Alexa Fluor 488 goat anti-rabbit and goat anti-mouse for anti-LC3II and anti-p62, respectively. The fluorescence images for intracellular LC3II (green) and p62 (green) accumulation, and iDyePSNPLs at emission wavelengths of 520 (green) and 585 nm (red), respectively, were captured using a Leica TCS SP5 confocal microscope. The Imaris 9.5 software was used for image processing. Chloroquine (100 µM), as a positive control, was added to the cells 2 h before the end of the 24 h period. Two independent experiments were conducted, and three or more fields were selected per sample, randomly. The quantification of the relative average area of fluorescence (arbitrary units: A.U.) was accomplished using ImageJ by randomly selecting three or more fields per sample.

### 2.9. Statistical Analysis

Statistical analyses were performed using GraphPad Prism 5 software. Three experiments were carried out by each one of the evaluated targets. Data were analyzed using unpaired student’s *t*-test and one-way ANOVA with Dunnett’s post-test. Statistical significance was defined as * *p* ≤ 0.05, ** *p* ≤ 0.01, and *** *p* ≤ 0.001.

## 3. Results and Discussion

### 3.1. Physicochemical Characterization of the Used PSNPLs

The TEM images show a mean size of 77.48 nm for PS-50, with a standard deviation of 11.13 nm, and a 446.28 nm average size with a calculated standard deviation of 10.04 nm for PS-500. For both cases, we observed an estimated average uniformity in the particle size distribution, which translated to a PDI of 0.02 and 0.01, respectively. Even considering that the individual size of the particles, and therefore the size distribution of the particles, was homogeneous, there is an indicator of the potential effects of the used dispersant (the Milli-Q water or the culture medium), as observed in Figure 1A–F for PS-50 and PS-500, respectively. As observed in Figure 1C,F, the culture medium induced some agglomeration but without affecting the size and, consequently, the size distribution. The TEM figures, especially for PS-500, show the effects of the culture medium inducing a corona containing different components of the culture medium. This corona could be responsible for the observed agglomeration. In order to confirm the role of the culture medium on the PS-50/PS-500 characteristics, their hydrodynamic behavior was evaluated by DLS. It was observed, in both cases, that a shift to the right occurred when they were suspended in the culture media instead of water (Figure 1G,I). For each measurement, the obtained correlations are indicated (Figure 1H,J). Remarkably, there was a clear shift in the Z-potential from −44.90 to −13.70 for PS-50 and from −50.40 to −11.30 when suspended in the culture media. This may reflect the influence of the culture media over the stability of the PSNPLs regardless of the obtained sizes, as also observed in the TEM images. Additionally, confirming the identity of polystyrene polymer, we can see all major functional groups represented on the FTIR diagram in Figure 1K. Peaks at the 3081, 3059, and 3025 regions are assigned to C-H stretching vibration absorption, while 2921 and 2850 are assigned to methylene. Additionally, 1600, 1492, and 1452 correspond to C=C aromatic vibration absorption [33], and 756.0 and 698.2 correspond to out-of-plane C-H bending vibration absorption and indicate that there was only one substituent in the benzene ring [34].

### 3.2. Cell Viability Assessment

Initially, the present study determined the potential decrease in cell viability of HNEpCs exposed to different concentrations of PS-50 and PS-500, ranging from 0 to 100 µg/mL, as an indicator of cytotoxicity. As depicted in Figure 2, there were no significant decreases in cell viability after exposures lasting for 24 h with PS-50 and PS-500 at any of the tested concentrations. When put together, this indicates that neither of the different sizes of PSNPLs were able to cause cytotoxicity after acute (24 h) exposure. Similarly, a wide range of PS-50 nm (25 to 200 µg/mL) concentrations also did not reveal any cytotoxic response after 24 h of exposure to the Caco-2/HT29 monolayers of both models and to the Caco-2/HT29 + Raji-B cells from the triculture model [8]. However, a 200 µg/mL concentration of PS-50 nm was able to cause significant cell death in blood cell types like Raji-B and TK6 cells after 24 and 48 h of exposure, possibly due to cell type-specific effects [2].

### 3.3. Intracellular Localization of PSNPLs

The uptake and the intracellular localization of iDyePS-50 and iDyePS-500 were determined in the cells exposed to 100 µg/mL in treatments lasting for 24 h by using confocal microscopy. As observed in Figure 3, iDyePS-50 and iDyePS-500 were detected inside the cells, mainly localized in the cytoplasm. This confirms the ability of the selected cells (HNEpCs) to internalize PSNPLs. Interestingly, a greater internalization of iDyePS-50 was observed in comparison with iDyePS-500, indicating a size-dependent effect under the conditions used in this study. The intracellular uptake of fluorescence PSNPLs with a diameter of 42 nm was also proven in HNEpCs [21]. The authors pointed out that the internalization was possibly due to phagocytosis and, as a consequence, the PSNPLs entering the nucleus, inhibiting cell proliferation and causing cell apoptosis and necrosis.

### 3.4. Generation of iROS by PSNPLs Exposure

The pristine forms of MNPLs could potentially cause excessive production of intracellular ROS in many mammalian cell line models upon cellular internalization. The free radicals thus generated could play a role in several oxidative stress response cascades in cells. The imbalance or higher incidence of iROS could be due to instability in the mitochondrial membrane potential (MMP) [35,36]. The dysfunction of the mitochondrial membrane could be indirectly affected by MNPLs due to increased ROS in cellular compartments outside of or nearby mitochondria [37]. Hence, the estimation of the iROS levels induced by PS-50 and PS-500 exposure in the HNEpCs was determined after exposures lasting for 24 h. As observed in Figure 4, the flow cytometry data show a significant increase in the percentage of intracellular ROS induction (as DCFH-DA fluorescence intensity) when HNEpCs were treated with 100 µg/mL of both PS-50 and PS-500, as compared to untreated cells (Figure 4A,B, *p* < 0.05). Nevertheless, no significant differences were observed between the sizes when the effects of PS-50 and PS-500 were compared; despite that, PS-50-treated cells had slightly elevated levels of iROS versus those induced by PS-500 (Figure 4B). These effects could be attributed to the cellular internalization and localization in different intracellular regions of the cells. In support, the exposure of BEAS-2B and alveolar (HPAEpiC) epithelial cells to PSNPLs (40 nm) resulted in a redox imbalance in the cells due to the higher production of iROS after 24 h, which could probably be involved in lung injury [19]. Besides, three different types of PSNPLs, namely amino-functionalized PS-NPL20, PS-NPL20, and PS-NPL50 nm, were able to induce a dose-dependent iROS induction and the increased expression of NADPH oxidase 4 (NOX4), which plays a critical role in the process of PSNPL-induced epithelial-to-mesenchymal transition in A549 cells [38].

### 3.5. Loss of Mitochondrial Membrane Potential by PSNPLs

The stability of mitochondrial membrane potential is paramount for maintaining the normal physiological functions of cells. However, the loss of its membrane potential could be associated with elevated levels of iROS, leading to mitochondrial damage [36]. Several recent studies with PSNPLs have shown potential mitochondrial damage either directly or indirectly by increased iROS [39,40,41]. Since mitochondria are potential targets of PSNPLs exposure, the present study investigated the effects of PS-50 and PS-500 on the mitochondrial membrane potential (MMP) as an indicator of mitochondrial damage in HNEpCs after exposures lasting 24 h. As indicated in Figure 5, the obtained data revealed a significant loss of mitochondrial membrane potential associated with PS-50 exposure. Besides, there was also a slight decrease in, or loss of, MMP observed when HNEpCs were exposed to PS-500. In both cases, the exposure effects were compared to those observed in the untreated control group (Figure 5A,B; *p* < 0.05). These results also suggested that the loss of MMP could be associated with the size of the used PSNPLs, where a smaller size (PS-50) had a more pronounced MMP effect than greater sizes (PS-500) in HNEpCs exposed for 24 h. In A549 cells, amino-functionalized PS-NPL20, PS-NPL20, and PS-NPL50 nm were able to affect mitochondrial function due to a loss of MMP and impaired cellular energy metabolism by proton leaks and the increased adenosine triphosphate (ATP) production of mitochondrial oxidative phosphorylation [38]. Further, different sizes of PS microplastics (0.3 to 6 µM) with and without the adsorption of bisphenol A (BPA) could induce iROS and decrease MMP based on the size and adsorption of BPA to the PS plastics in Caco-2 cells as a toxic response [42]. Furthermore, other authors also confirmed a reduced MMP due to pristine and amino-functionalized PS (50 nm) nanoplastics in HUVEC after 24 h of treatment [40]. Similarly, PS microplastics (5 µM) were involved in the loss of MMP and the destruction of the mitochondrial membrane structure and their genomic integrity, as well as an imbalance in homeostasis between the division and fusion of mitochondria in mouse spermatocyte (GC-2) cells. Moreover, it triggered mitochondrial autophagy via PINK1/Parkin-mediated pathway [41].

### 3.6. PSNPLs Cause Defective Autophagy

Exposure to environmental pollutants could lead to the induction of autophagy as a protective cellular strategy. Severe damage induction could lead to the loss of autophagic flux or insufficient autophagy, eventually contributing to cell death [43]. Recently, several studies demonstrated that PSNPLs could be involved in the modulation of the autophagy process by inhibiting or blocking it [44,45,46]. In the present study, the ability of PS-50 and PS-500 to cause defective autophagy or the loss of autophagy flux in the treated cells after 24 h was determined by studying two important autophagy markers: LC3-II and p62 using Western blotting (Figure 6). The upregulation of LC3-II protein expression could indicate an increase in, or the accumulation of, the intracellular autophagosomes, and p62, as a substrate for autophagy, would become degraded in a normal complete autophagic process. Nevertheless, impairment to the degradation of this ubiquitous protein could suggest the blockage of the pathway [47]. Our data demonstrated that PS-50 and PS-500 in the presence of chloroquine, an inhibitor of autophagosome and lysosomal fusion, were able to significantly increase the expression of the LC3-II marker in the treated cells, compared to the untreated control (Figure 6A, *p* < 0.05), although PS-50 and PS-500 alone did not show increases in the expression of LC3-II. Nevertheless, PS-50 and PS-500 alone, or in the presence of chloroquine, did show a significant change in the expression levels of the p62 marker in comparison to untreated controls (Figure 6B, *p* < 0.05). When taken together, these results suggested the potential role of PSNPLs in blocking the autophagy pathway, as evidenced by the accumulation of LC3-II and p62 markers in the treated cells.

Furthermore, we aimed to study the colocalization of iDyePS-50 and iDyePS-500 with the LC3-II and p62 biomarkers in the treated cells under confocal microscopy (Figure 7 and Figure 8). The obtained data indicated a significant accumulation of LC3-II in the presence of iDyePS-50 and iDyePS-500 in the cells after 24 h of treatment (Figure 7B,C,E). We observed LC3-II and iDyePS-50 nm (Figure 7B), as well as LC3-II and iDyePS-500 nm (Figure 7C) near each other in the cytoplasmic regions of the cells, indicating they could be involved in the accumulation of autophagosomes.

Interestingly, we also found significantly increased p62 puncta in the iDyePS-50- and iDyePS-500-treated cells compared to the untreated control (Figure 8B,C,E). There was a slight increase in p62 puncta in iDyePS-50 over iDyePS-500; however, this was not significant (Figure 8E). Moreover, there was also colocalization of iDyePS-50 and iDyePS-500, with accumulated p62 (inset in Figure 8B,C), suggesting insufficient autophagy due to the PSNPLs in the HNEpCs. This further corroborated that PS-50 and PS-500 could block the autophagy pathway upon cellular internalization.

In agreement with this, elevated levels of LC3-II and p62 were also observed in the PSNPLs (30 nm)-exposed macrophages (RAW246.7), indicating disrupted autophagic flux [48]. The pristine form of the PSNPLs (<100 nm) also pronounced significant increases in the expression of LC3-II in BEAS-2B cells in a dose-dependent manner [49]. Likewise, the PSNPLs caused a size-dependent accumulation of autophagosomes and a decrease in autolysosomes after exposure to HUVECs lasting 48 h, indicating an impairment in autophagic flux. The PSNPLs (100 nm) also showed significantly elevated levels of LC3-II, mCherry-GFP-LC3 puncta, and Beclin-1 in a dose-dependent manner; however, the 500 nm PSNPLs did not result in similar autophagic effects when compared. These differential effects could be attributed to the insignificant cellular internalization of the 500 nm PSNPLs [28].

Furthermore, the colocalization of LC3-II and the PSNPLs (100 nm) was also observed in mouse embryonic fibroblast cells, suggesting that autophagosomes contained PSNPLs [44]. Additionally, the PSMPLs (2 µM) exhibited an increased expression of LC3-II and Beclin-1 in a concentration-dependent manner. However, the expression of p62 was not significant in human kidney proximal tubular epithelial cells (HK-2 cells) [46], and exposure to different-sized PS microplastics (0.5, 1, and 5 μM) to HUVECs resulted in autophagic-mediated cell death [45].

## 4. Conclusions

The hazard assessment of MNPLs, such as PSNPLs, which are considered airborne due to their ever-increasing usage, is the need of the hour [50]. Since there are only very limited studies that focus on the respiratory system and on the upper respiratory region, our study involving HNEpCs deserves special relevance. Our present study attempted to show the cellular effects of PNSPLs of varying sizes in HNEpCs after their successful internalization into the cells. The intracellular presence of PSNPLs led to the excessive induction of iROS, the loss of MMP, and disturbances to the autophagy pathway. Notably, the blockage of autophagy due to PSNPLs is an interesting tipping point in elucidating and furthering the autophagy-related cellular effects. The implications of defective autophagy must be noted because PSNPLs are manifold at the cellular level. One of such implications is their potential role in the induction of cellular transformational effects, which cannot be discounted when considering the continuously increasing exposure to PSNPLs due to the ubiquitous environmental presence of MNPLs. In summary, we found that both PS-50 and PS-500 have pronounced and similar cellular responses in HNEpCs, albeit PS-50 was slightly more responsive, possibly due to its size.

## Figures and Tables

**Figure 1 biomolecules-13-00220-f001:**
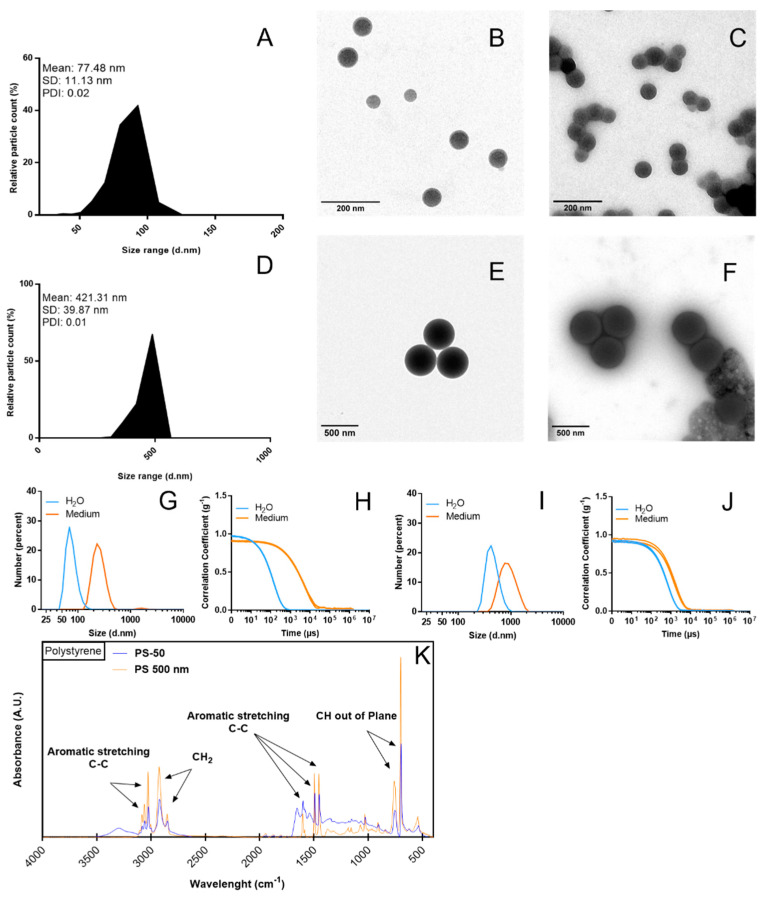
Particle size distribution (number percent) in Milli-Q water dispersion and the culture medium can be observed for both PS-50 (**A**–**C**) and PS-500 (**D**–**F**). The changes in hydrodynamic size and the correlation of the measurements are detailed for PS-50 (**G**,**H**) and PS-500 (**I**,**J**). The chemical composition of PS-50 (blue curve) and PS-500 (orange curve) shows the major functional groups measured by FTIR (**K**). d.nm, diameter in nm. PDI, polydispersity index.

**Figure 2 biomolecules-13-00220-f002:**
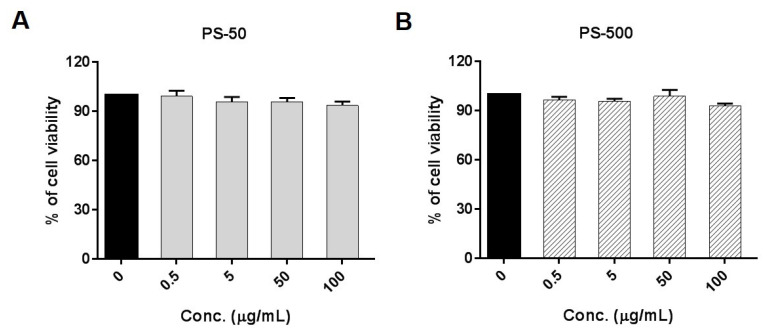
Polystyrene nanoplastic cytotoxicity in HNEpCs. Cell viability for PS-50 (**A**) and PS-500 (**B**) is indicated, showing the effects of increasing doses of PSNPLs for 24 h. Data are presented as the number of exposed cells relative to the nonexposed control ± SEM. Conc., concentration.

**Figure 3 biomolecules-13-00220-f003:**
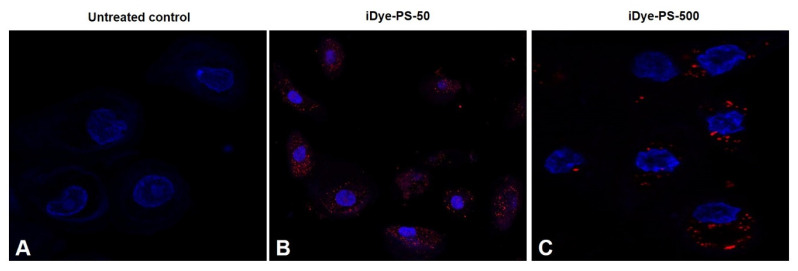
The confocal images depict the internalization of iDyePS-50 and iDyePS-500 in HNEpCs after exposures (100 µg/mL) lasting for 24 h. (**A**) Untreated control. (**B**) Localization of iDyePS-50 (in red) in cytoplasmic regions and surrounding nuclei (blue). (**C**) Presence of iDyePS-500 (in red) in the cytoplasm and proximity to nuclei.

**Figure 4 biomolecules-13-00220-f004:**
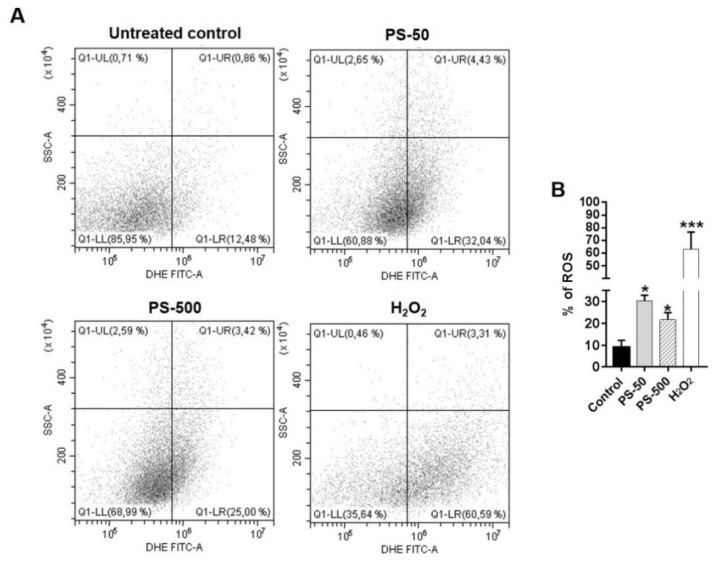
The dot plots and graph indicated the generation of intracellular reactive oxygen species (iROS) due to PSNPLs in HNEpCs after 24 h of treatment. (**A**) The dot plots represented the increased percentage of iROS in both PS-50- and PS-500-treated cells (far right quadrant) over the untreated control. H_2_O_2_ acted as a positive control. (**B**) The graph represented the significant increase in the percentage of iROS due to PS-50 and PS-500 in the treated cells vs. untreated control cells. * *p* ≤ 0.05, *** *p* ≤ 0.001.

**Figure 5 biomolecules-13-00220-f005:**
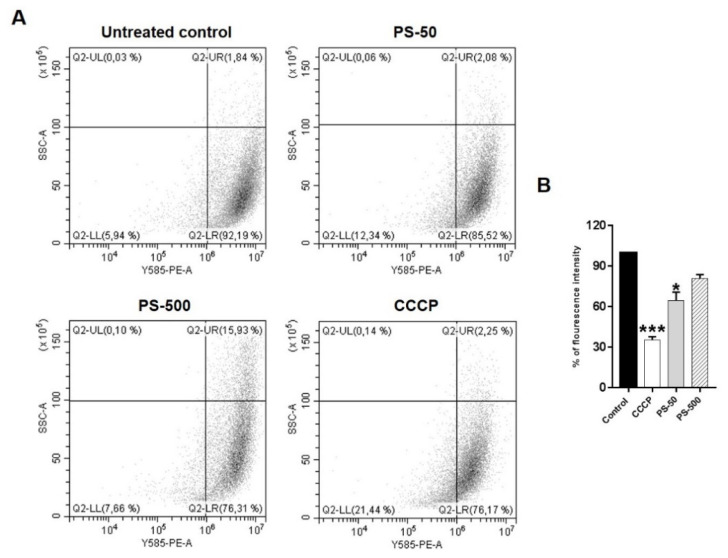
The dot plots and graph indicate the loss of mitochondrial membrane potential due to PSNPL exposure in HNEpCs after 24 h of treatments. (**A**) The dot plots represented the decrease in fluorescence intensity as an indicator of loss of MMP due to treatment of HNEpCs with PSNPLs (far left quadrant) over the untreated control. CCCP acted as a positive control. (**B**) The graph represents the significant loss of MMP due to PS-50 in the treated cells vs. untreated control cells. * *p* ≤ 0.05, *** *p* ≤ 0.001.

**Figure 6 biomolecules-13-00220-f006:**
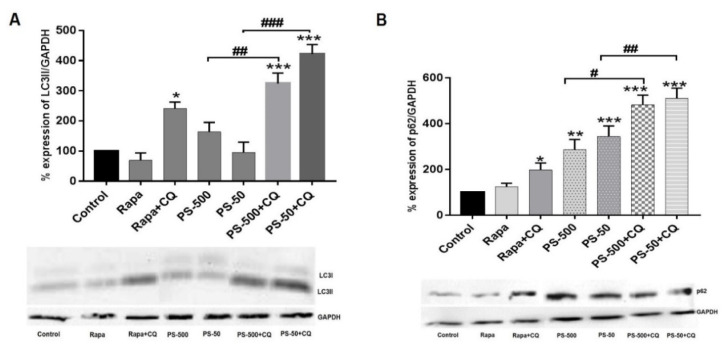
Images showing the expression of important autophagy pathway proteins: LC3-II and p62 after exposure of HNEpCs to PS-50 and PS-500 alone or their combination with chloroquine for 24 h. (**A**) Histogram showing the quantitative changes in the levels of LC3-II expression in treated and untreated cells (top) and the respective Western blot for LC3-II (bottom). (**B**) Histogram showing the quantitative changes in the levels of p62 expression in the treated and untreated cells (top) and respective Western blot for p62 (bottom). * *p* ≤ 0.05, ** *p* ≤ 0.01, *** *p* ≤ 0.001 vs. untreated control; ^#^
*p* ≤ 0.05, ^##^
*p* ≤ 0.01, ^###^
*p* ≤ 0.001 (PS-50/PS-500 vs. PS-50/PS-500 + CQ). Note: Rapa and CQ in the graphs and blots represent rapamycin and chloroquine, respectively.

**Figure 7 biomolecules-13-00220-f007:**
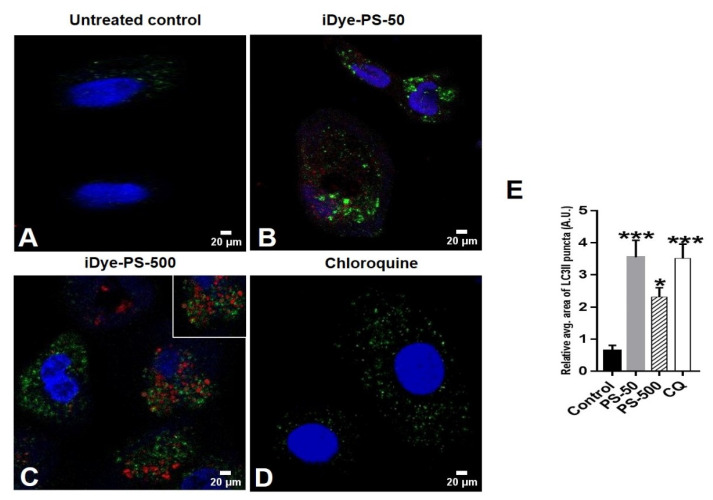
The confocal images represent the accumulation of LC3II (green) after exposure of HNEpCs to 100 µg/mL of iDyePS-50 (red) and iDyePS-500 (red) for 24 h. (**A**) Untreated control. (**B**) Concurrent presence of iDyePS-50 (red) and LC3II puncta (green) in cytoplasmic regions. (**C**) Overlapping of iDyePS-500 (red) and LC3II (green) in cytoplasmic regions or colocalization of iDye-PS500 and LC3II (yellow, shown in inset). (**D**) chloroquine as a positive control for LC3II (green). (**E**) The histogram depicts the measurement of the relative average area of fluorescence (green/LC3II puncta, AU) due to iDyePS-50 and iDyePS-500 over the untreated control. * *p* ≤ 0.05, *** *p* ≤ 0.001 vs. untreated control.

**Figure 8 biomolecules-13-00220-f008:**
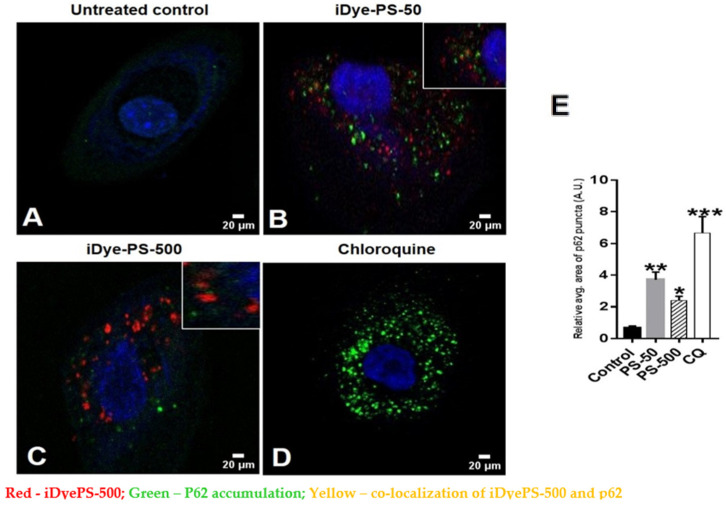
The confocal images represent the expression of p62 (green) after exposure of HNEpCs to 100 µg/mL of iDyePS-50 (red) and iDyePS-500 (red) for 24 h. (**A**) Untreated control. (**B**) Observation of iDyePS-50 (red) and p62 (green) puncta together in cytoplasmic regions or colocalization of iDyePS-50 and p62 (yellow, depicted in inset). (**C**) Presence of iDyePS-500 (red) and p62 (green) together in cytoplasmic regions or colocalization of iDyePS-500 and p62 (yellow, depicted in inset). (**D**) chloroquine as a positive control for p62 (green). (**E**) The histogram depicts the measurement of the relative average area of fluorescence (green/p62 puncta, AU) due to iDyePS-50 and iDyePS-500 over the untreated control. * *p* ≤ 0.05, ** *p* ≤ 0.01, *** *p* ≤ 0.001 vs. untreated control.

## Data Availability

Data is available under request.

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
