# Peer review of "Hazard Assessment of Polystyrene Nanoplastics in Primary Human Nasal Epithelial Cells, Focusing on the Autophagic Effects"

_biomolecules, 2023, doi:10.3390/biom13020220_

Round 1
Reviewer 1 Report
The authors have done a study on the interaction of PS nanoplastics to human alveolar cells.
THe materical characterization part is missing.
The authors should have done in real dust samples or SRM dust laced with PS nanoplastics and done it.
In the alveolar cells the PS nanoplastics will react as such. It combines with other molecules in their way. THe reality of the issue is missing.
Author Response
see the enclosed document

Reviewer 2 Report
The main goal of this project was to assess the potential hazard posed by polystyrene nanoplastiscs (PSNPLs of 50 nm and 500 nm) in human primary nasal epithelial cells (HNEpCs). In my opinion, this work is rather a routine cytotoxicity study, which describes the results of oxidative stress, mitochondrial membrane potential, and autophagy assays in HNEpCs cell, after acute exposure (24 hour) to commercially purchased PSNPLs. However, the amount of work is extensive and the relevance of the PSNPLs as environmental pollutants might be worth consideration for publication after addressing the following major and minor comments.
1) The experiment design does not seem to be well justified.
For example, cell exposure with PSNPLs was in the 0.50 to 100 µg mL range. How were these PSNPL concentrations selected? What is their environmental and human health relevance with respect to potential human inhalation? Why PSNPLs of 50 nm and 500 nm? In my opinion, two different PSNPL sizes are not representative of a size-dependent study as claimed by the authors (see below citation). Also, the selection of a single exposure time, which was defined as acute exposure, must be carefully explained.
On page 3, lines #103 and 104:
“As we were interested in the size’s effects, two pristine PSNPLs (50 and 500 nm) were selected.”
2) The physicochemical characterization of the PSNPLs seems very modest.
For example, panel k in figure 1 presents the FT-IR spectrum of both PSNPLs (50 nm and 500 nm). The graph is not readable, and it is not clear what we learn from the labeled peaks. The authors should consider describing this chemical characterization in the actual text of the manuscript.
It is not clear what is the increase in the PSNPL diameters because of their incubation with the cell culture medium. What are these DLS values and how do they compare to the TEM values? Why was a protein corona observed only for the PSNPLs of 500 nm in diameter but not the ones of 50 nm?
On page 5, lines#241-242:
“As observed, the hydrodynamic size increases for both PS-50/PS-500 when particles were dispersed in the culture medium (g, i).”
3) The bioassay results of this study are not well correlated with those presented from literature for comparative purposes.
For example, it is not clear to me how the “no effects” observed for the exposure with PSNPL of 500 nm correlates with the presented examples from literature on mitochondrial membrane potential. Was this size related trend observed by others or not? If yes, what was the mechanism behind this size-dependency? If not, a potential explanation for the observed trends for these two specific PSNPL sizes (50 nm versus 500 nm) should be presented.
On page 9, lines #325-327:
“As indicated in Figure 5, the obtained data revealed a significant loss of mitochondrial membrane potential associated with PS-50 exposure. Nevertheless, no effects were observed when HNEpCs were exposed to PS-500.”
4) The manuscript needs careful editing.
For example, space should be inserted between a parameter value and its corresponding unit. End-of-line hyphenation and grammar must be revised. All confocal images (e.g., figures 3, 7, and 8) lack scale bars.
On page 4, line #160: “585nm”
On page 4, lines #161-162: “Two independ-
ent experiments were conducted and two random fields were selected per sample.”
Author Response
see the attached document

Reviewer 3 Report
Manuscript ID-biomolecules-2131633
In this research focussed on the potential hazard of polystyrene nanoplastics (PSNPLs with sizes 50 and 500 nm) in human primary nasal epithelial cells (HNEpCs). MS is compact and focuses on the problem taken up.
MS is well written and no issues with the English language or grammatical mistakes. I recommend for its publication after the incorporation of few minor corrections/suggestions:
1) Line 34 (Introduction section): Since this study focuses on PSNPLs with sizes 50 and 500 nm, In introduction section, I recommend that authors should emphasize the significance of studying PSNPLs at different scales (especially nanoscale plastics). Why did author focuses with 50 and 500 nm?
2) Line 131,132.: References in context, please try to follow a similar pattern throughout the MS as well as in (Ramezanpour et al., 2018, Scherzad et al., 2019). The authors should be insert ‘;’ after 2018.
3) Line 231. Explain the term "PDI". I suggest to explain it before results & discussion.
4) Line 260: Please, change ‘(I, j)’ for ‘(i, j)’, it should be in small letters.
5) Line 384: ‘iDye PS-500,’ Will it be ‘iDyePS-500,’ Please check space bar.
6) Line 285-305 (Section 3.4): The authors listed a lot of similar researches as shown in section 3.4. But how much concentration of PSNPLs that were treated in those references?
7) Line 434 (Conclusion section): PS-50 and PS-500, Which is strong hazard for NHEpCs? It should be written based on this finding.
Author Response
see the attached document

Round 2
Reviewer 2 Report
I would like to thank the authors for considering the suggestions for improvement and their efforts. All comments except for the one restated below have been addressed:
"In my opinion, two different PSNPL sizes are not representative of a size-dependent study. "
Author Response
We moved through the Title, Abstract, and Conclusions, and in any of the cases, we emphasized that this was a size-dependent study. Nevertheless, to avoid misinterpretations, we have included at the end of the Introduction section, the following statement: “Nevertheless, we are aware that this is not a complete size-dependent study, since only two sizes were evaluated”.